# Effective Treatment against ESBL-Producing *Klebsiella pneumoniae* through Synergism of the Photodynamic Activity of Re (I) Compounds with Beta-Lactams

**DOI:** 10.3390/pharmaceutics13111889

**Published:** 2021-11-08

**Authors:** Iván A. González, Annegrett Palavecino, Constanza Núñez, Paulina Dreyse, Felipe Melo-González, Susan M. Bueno, Christian Erick Palavecino

**Affiliations:** 1Departamento de Química, Facultad de Ciencias Naturales, Matemática y del Medio Ambiente, Universidad Tecnológica Metropolitana, Las Palmeras 3360, Ñuñoa, Santiago 7800003, Chile; igonzalezp@utem.cl; 2Laboratorio de Microbiología Celular, Instituto de Investigación e Innovación en Salud, Facultad de Ciencias de la Salud, Universidad Central de Chile, Lord Cochrane 418, Santiago 8330546, Chile; annegrett.palavecino@alumnos.ucentral.cl (A.P.); constanza.nunezc@alumnos.ucentral.cl (C.N.); 3Departamento de Química, Universidad Técnica Federico Santa María, Av. España 1680, Casilla, Valparaíso 2390123, Chile; paulina.dreyse@usm.cl; 4Departamento de Genética Molecular y Microbiología, Millennium Institute on Immunology and Immunotherapy, Facultad de Ciencias Biológicas, Pontificia Universidad Católica de Chile, Santiago 8330025, Chile; famelo@bio.puc.cl (F.M.-G.); sbueno@bio.puc.cl (S.M.B.)

**Keywords:** photodynamic therapy, multi-drug resistance, antibiotic synergy, *Klebsiella pneumoniae*

## Abstract

Background: Extended-spectrum beta-lactamase (ESBL) and carbapenemase (KPC^+^) producing *Klebsiella pneumoniae* are multidrug-resistant bacteria (MDR) with the highest risk to human health. The significant reduction of new antibiotics development can be overcome by complementing with alternative therapies, such as antimicrobial photodynamic therapy (aPDI). Through photosensitizer (PS) compounds, aPDI produces local oxidative stress-activated by light (photooxidative stress), nonspecifically killing bacteria. Methodology: Bimetallic Re(I)-based compounds, PSRe-µL1 and PSRe-µL2, were tested in aPDI and compared with a Ru(II)-based PS positive control. The ability of PSRe-µL1 and PSRe-µL2 to inhibit *K. pneumoniae* was evaluated under a photon flux of 17 µW/cm^2^. In addition, an improved aPDI effect with imipenem on KPC^+^ bacteria and a synergistic effect with cefotaxime on ESBL producers of a collection of 118 clinical isolates of *K. pneumoniae* was determined. Furthermore, trypan blue exclusion assays determined the PS cytotoxicity on mammalian cells. Results: At a minimum dose of 4 µg/mL, both the PSRe-µL1 and PSRe-µL2 significantly inhibited in 3log_10_ (>99.9%) the bacterial growth and showed a lethality of 60 and 30 min of light exposure, respectively. Furthermore, they were active on clinical isolates of *K. pneumoniae* at 3–6 log_10_. Additionally, a remarkably increased effectiveness of aPDI was observed over KPC^+^ bacteria when mixed with imipenem, and a synergistic effect from 3 to 6log_10_ over ESBL producers of *K. pneumoniae* clinic isolates when mixed with cefotaxime was determined for both PSs. Furthermore, the compounds show no dark toxicity and low light-dependent toxicity in vitro to mammalian HEp-2 and HEK293 cells. Conclusion: Compounds PSRe-µL1 and PSRe-µL2 produce an effective and synergistic aPDI effect on KPC^+^, ESBL, and clinical isolates of *K. pneumoniae* and have low cytotoxicity in mammalian cells.

## 1. Introduction

Due to the emergence of multi-drug resistance (MDR) pathogenic bacteria, the deficit of new antibiotics is one of the most pressing threats to human health in the 21st century [1]. The world health organization has presented a ranking of the most relevant MDR bacteria that require the urgent development of new antimicrobial therapies. Strains of *Klebsiella pneumoniae* producing extended-spectrum β-lactamase (ESBL) and carbapenemase (KPC) are among the most relevant [2,3]. *K. pneumoniae* is a Gram-negative bacillus associated with pneumonia and urinary tract infections (UTI) [4,5]. Additionally, *K. pneumoniae* is one of the most relevant agents of healthcare-associated infections (HAIs) [6]. The HAIs produced by *K. pneumoniae* can be severe, producing mortalities as high as 30 to 70% [7,8]. The use of polymyxins (colistin) or tigecycline antibiotics are the only therapeutic options to treat severe KPC^+^ infections [9]. The global increase in pan-resistant Enterobacteriaceae has resulted in increased use of colistin, which has accelerated the onset of resistance to polymyxins; the emergence of the mcr-1 gene is a good example [1,10]. Therefore, MDR-*K. pneumoniae* strains have a great potential to become a “superbug”; therefore, they are an excellent model for discovering new antimicrobial treatments [8].

In this scenario, non-antibiotic therapeutic options with antimicrobial properties should be explored. An alternative is the antimicrobial photodynamic inactivation (aPDI) based on light-activated photosensitizer compounds (PS) [11,12,13]. The PSs are chemical compounds that absorb and accumulate the quantized energy of a specific wavelength accessing a triplet excited state by intersystem crossing processes [14]. The accumulated energy is transferred to the molecular oxygen commonly present in biological solutions through two mechanisms of action to produce reactive oxygen species (ROS): The Type I effect transfers energetic electrons that produce superoxide (O_2_**^•−^**); the O_2_**^•−^** produces other ROS, such as hydrogen peroxide (H_2_O_2_) and hydroxyl radical (HO**^•^**) [15,16]. The Type II effect transfers the energy (with no electrons) to generate singlet oxygen (^1^O_2_) [17,18]. ROS such as ^1^O_2_ produces photooxidative stress by concerted addition reactions of alkene groups in closer organic macromolecules such as protein alkylation, lipid carboxylation, and DNA degradation [12,19,20]. Hence, photooxidative stress results in non-specific bacterial cell death produced by damage over bacterial structures such as plasma membranes or DNA [17,21]. Many initiatives have developed PS compounds with aPDI properties for bacteria such as *K. pneumoniae* [13,20,22,23,24]. PS with a longer-lived excited state lifetime improve the probability of interacting with triplet oxygen and must produce more ^1^O_2_ [25,26,27,28]. Moreover, it has been identified that cationic PSs produce more significant inhibition in bacterial growth [29], probably due to a more intimate interaction with the negatively charged bacterial envelope [30]. Therefore, cationic PSs may show a better photodynamic effect on *K. pneumoniae* than anionic PSs [29,30,31]. Our laboratory has tested various cationic Ir(III) organometallic PSs with antimicrobial properties against *K. pneumoniae* [23,29,32]. Other authors have also developed PS for *K. pneumoniae* [31,33,34,35], where some of these are organic molecules able to inhibit bacterial growth in vitro [33]. The PS molecule must show low levels of cytotoxicity to reduce the probability of adverse pharmacological effects. In this regard, the PSs based on organic molecules should be less toxic [19]. Other coordination compounds based on transition metals into tetrapyrrole structures or 5-aminolevulinic acids (ALA) have been successfully used for photodynamic treatment of cancer [36,37]. The Ru(II) complex containing three phenanthroline ligands has shown highlighted antimicrobial activity [38]. Additionally, the Re(I) complexes have been used in vitro against a broad spectrum of bacteria [39]. In this sense, the complexes with transition metals such as Re(I) can be considered good options according to their photophysical properties to produce reactive oxygen species useful for antimicrobial treatment [40].

Here we verify that PS compounds that absorb in a wide range of the visible spectrum, such as bimetallic Re (I) bimetallic complexes with polypyridine bridging ligand, may be helpful in aPDI over *K. pneumoniae*. These complexes have (Re(CO)_3_Cl)_2_μ-N^N general formula and here were evaluated with the following N^N ligands: 2,3-Dicarboxypyrazino [2,3-f] [4,7] phenanthrolinedicarboxylic (L1) and 2,3-Diethoxycarbonylypyrazino [2,3-f] [4,7] phenanthroline (L2) to obtain the corresponding PSRe-µL1 and PSRe-µL2 compounds [41]. The photodynamic effect of the PSRe-µL1 and PSRe-µL2 compounds was tested in vitro in two laboratory strains of *K. pneumoniae*: the carbapenem susceptible (KPC**^−^**) KPPR-1 and the carbapenem-resistant (KPC^+^) ST258 strain, and also in a previously characterized population of 118 clinical isolates of *K. pneumoniae*, including 66 ESBL-producers [42]. In addition, the PSs capacity to inhibit bacterial growth was verified, as well as the pharmacological properties, such as minimum effective dose (MEC), lethality time, and synergy with imipenem (Imp) and cefotaxime (Cfx) antibiotics. Finally, the low cytotoxicity in mammalian cells determined in vitro makes these PSs a promising alternative to complement the treatment of complicated infections.

## 2. Materials and Methods

### 2.1. Synthesis of the Photosensitizer Compounds

Our group had previously synthesized and characterized the structure, photophysics, and purity of the PSRe-µL1 and PSRe-µL2 compounds, published in González et al., 2020 [41]. The characterizations included nuclear magnetic resonance (NMR), Fourier transforms infrared spectroscopy (FT-IR), elemental analysis TD-DFT calculations, and cyclic voltammetry. Additionally, the absorption spectra measured in acetonitrile solution were performed in a Shimadzu UV–Vis-NIR 3101-PC spectrophotometer. Finally, the molar extinction coefficients of the characteristic absorption bands and the area under the curve between 500–700 nm were determined. The complexes can be described using the following general formula: (Re(CO)_3_Cl)_2_µ-N^N, where N^N is 2,3-Dicarboxypyrazino [2,3-f] [4,7] phenanthrolinedicarboxylic (L1) or 2,3-Diethoxycarbonylypyrazino [2,3-f] [4,7] phenanthroline (L2). The Re(I) PSs obtained with both L1 and L2 are then designated as PSRe-µL1 and PSRe-µL2, respectively.

### 2.2. Antimicrobial Activity of Photosensitizers Compounds

Stock solutions of 2 mg/mL of each photosensitizer compound were prepared in dimethyl sulfoxide DMSO (Sigma-Aldrich, Saint louis, MO, USA), from which suitable work concentrations were obtained by dilution in an aqueous medium. The imipenem susceptible (KPC-) strain KPPR1 and the imipenem resistant strain ST258 (KPC^+^) of *K. pneumoniae* were used for the antimicrobial assay. Additionally, 118 clinical isolates of *K. pneumoniae* were used. Those clinical isolates were previously characterized for sensitivity and pathogenicity patterns [42]. For photodynamic experiments, all bacteria were grown as axenic culture in Luria Bertani medium and suspended at 1 × 10^7^ colony forming units (CFU)/mL cation-adjusted Muller Hinton broth (ca-MH). Bacteria were mixed with each PS in 24-well plates, in a final volume of 500 µL and light-irradiated immediately after adding the PS in a chamber with a white LED lamp at a photon flux of 17 µW/cm^2^. Controls plates with bacteria but no PS and with PS and no light were also included. All plates, including controls, were incubated for 1 h or the indicated time, and broth-micro dilution and sub-cultured on ca-MH agar plates were used to determine the CFUs of the viable bacteria. The agar plates were incubated at 37 °C, and colony count was recorded using a stereoscopic microscope after 16–20 h of incubation in the dark, following the recommendations of the Clinical and Laboratory Standards Institute (CLSI 2017) [43]. Control wells with bacteria, with or without photosensitizer but not exposed to light, were also included.

### 2.3. Determination of the Synergy between PSs and Cfx

To determine the fractional inhibitory concentration index (FIC) value, the following formula was used [44,45]:$$FIC\;Index=\frac{MICac}{MIC_{a}}+\frac{MICbc}{MIC_{b}}\;$$

MICac is the MIC of compound A, combined with compound B, and MICbc is the MIC of compound B combined with compound A. The MICa and MICb are the MIC of the A and B compounds alone, respectively. Values in the FIC index ≤ 0.5 are considered synergistic, and values > 4 are considered antagonistic [44]. To determine the MIC- Cfx in combination with each PS, 1 × 10^7^ UFC/mL of ESBL-producing bacteria were aPDI treated for 30 min with 4 µg/mL of each PSRe mixed with serial dilution (32–0.125 µg/mL) of Cfx in ca-MH broth, as stated above. To determine the modification in the PS-MEC when combined with Cfx, 1 × 10^7^ UFC/mL of ESBL-producing bacteria were added to 4 µg/mL of Cfx and mixed with a serial dilution of each PS (32–0.125 µg/mL) in ca-MH broth, as mentioned previously.

### 2.4. Cell Culture

The human cell lines from the American-type culture collection (ATCC), HEp-2 (CCL-23), and HEK293T (CRL-3216) were grown in DMEM with a mix of 1% streptomycin/penicillin antibiotics. The medium was supplemented with 10% fetal bovine serum (FBS), and the cultures were incubated in a 5% CO_2_ atmosphere. Initial cultures of 5 × 10^5^ cells per well were incubated for 24 to 48 h to a 70–90% confluence in triplicate in 24-well plates.

### 2.5. Cytotoxicity Tests in Eukaryotic Cells

The cytotoxic effect of PS compounds on HEp-2 and HEK293T cells was determined by evaluating cell death by the exclusion of trypan blue. The 24-well plates with cells at 70–90% confluence were mixed with 4 µg/mL of PSRe-µL1 and PSRe-µL2 compounds or control with D-PBS and incubated for 1 h in the dark or light-activated in a white LED light chamber with 17 µW/ cm^2^ of photon flux. Subsequently, the PSs were removed by washing twice with D-PBS, and the cells were incubated in the dark for an additional 24 h in DMEM supplemented with antibiotics + 10% FBS in a 5% CO_2_ atmosphere. After incubation, the cells were trypsinized, and the cell death was determined by trypan blue exclusion in a hemocytometer chamber.

### 2.6. Statistical Analysis

The Prism version 9.0 Software (GraphPad Software, LLC, San Diego, CA, USA) was used to perform the statistical analysis. Statistical significance was assessed using a two-tail *T*-test for parametric pairing groups or one-way ANOVA followed by the Tukey post-test for the lethality curves.

## 3. Results

### 3.1. Absorption Properties of the PSRe-µL1 and PSRe-µL2 Compounds

We previously showed that cationic Ir(III) are effective photosensitizers against *K. pneumoniae* [29]. However, these compounds present absorption peaks below 400 nm, limiting their excitation with wavelengths that better penetrate tissues [11]. Two bimetallic Re (I) compounds have been tested to enhance the PS activation in deeper tissue infections. Those bimetallic Re(I) complexes present polypyridine ligands (Figure 1A,B), they are fully structurally characterized, and they are thermodynamically stable in acetonitrile solution [41,46]. As shown in Table 1 and Figure 1D, the PSRe-µL1 and PSRe-µL2 present similar absorption processes at 358 and 360 nm, respectively. However, they show dark coloration attributed to a wide range of absorption bands in the visible range, with the maximums at 587 and 588 nm (Figure 1C,D). As shown in Figure 1D, both PSRe-µL1 and PSRe-µL2 compounds have an intense dark color, which shows their characteristic of absorbing light with a significant molar extinction coefficient in a wide visible range. The broad range absorption (500–700 nm) shows a significant area under the curve of 16.28 and 14.39 for PSRe-µL1 and PSRe-µL2, respectively. The compounds also present significantly high molar absorption of 3384 and 5600 M^−1^cm^−1^ for the PSRe-µL1 and PSRe-µL2, respectively (Table 1). The absorptions are attributable to the singlet metal-to-ligand (^1^MLCT) and singlet ligand-to-ligand (^1^LLCT) charge transfer transitions from the metal and chloride ligand to ligand bridge (L1 or L2), following the behavior of analogous compounds and the trends of cyclic voltammetry experiments and TD-DFT calculations [41]. In addition, the cyclic voltammetry showed a first oxidation process at 1.51 and 1.70 for PSRe-µL1 and PSRe-µL2, respectively. A second oxidation process was obtained only for PSRe-µL1 at 1.65 V (Table 1). The HOMO-LUMO energy gaps calculated for the PSRe-µL1 and PSRe-µL2 complexes were of similar magnitude, 2.73 and 2.84 eV, respectively (Table 1).

The PS-Ru [Ru(bpy)_3_](PF_6_)_2_ (bpy = 2,2′-bipyridine) compound was used as a control to compare the aPDI activity of the PSRe-µL1 and PSRe-µL2 compounds [38]. According to the literature, in acetonitrile, the PS-Ru shows a charge-transfer absorption process at 450 nm [48] and a significantly low molar extinction coefficient at 550 nm of ~600 M^−1^cm^−1^ [47]. In addition, we determine its area under the curve of 0.83 between 500–700 nm wavelength, which is significantly smaller than the PS-Re compounds (Table 1).

### 3.2. Antimicrobial Photodynamic Inactivation Activity of Compounds PSRe-µL1 and PSRe-µL2

The photodynamic antimicrobial capacity of the two new Re (I) compounds was determined by inhibiting the bacterial growth of *K. pneumoniae*. PSRe-µL1 and PSRe-µL2 at 16 µg/mL were then mixed with each *K. pneumoniae* strain, KPPR1 (KPC^−^) and ST258 (KPC^+^) at 1 × 10^7^ CFU/ml. The aPDI activity of the PSRe-µL1 and PSRe-µL2 compounds was compared with the reference PS-Ru antimicrobial activity as a positive control [34,38,49]. Initial tests were performed with 16 µg/mL of each compound in an aqueous solution (ca-MH). Figure 2 shows the photodynamic treatment with PSRe-µL1 and PSRe-µL2 (Green bars), which inhibits in 3 log_10_ (<99.9%) the growth of both strains of *K. pneumoniae* (* *p* < 0.05) compared to untreated control bacteria (Blue bars). The results show that the bactericidal effect produced by both PSs is dependent on light (Red bars) (ns *p* > 0.05, compared to untreated control) because growth inhibition is observed after light activation. Therefore, the PSRe-µL1 and the PSRe-µL2 compounds must be activated by light to exhibit their bactericidal effect. Comparable results were obtained with the 16 µg/mL PS-Ru control, as the bacterial growth inhibition was observed only after light activation (*p* < 0.05).

### 3.3. Determination of the Minimum Effective Concentration and Lethality for the Compounds PSRe-µL1 and PSRe-µL2

Because the PSRe-µL1 and PSRe-µL2 compounds showed a significant aPDI effect on the two *K. pneumoniae* strains, a further two pharmacologic parameters were determined: the minimum effective concentration (MEC) and the minimum light exposition time (lethality) on the KPPR1 and the ST258 *K. pneumoniae* strains. The PS-MEC was determined by exposition of 1 × 10^7^ CFU/mL of each *K. pneumoniae* strain to serial dilutions (ranging from 0 to 32 µg/mL) of each PS in ca-MH broth. The aPDI treatment was performed for 1 h at a fluence rate of 17 µW/cm^2^. After treatment, viable bacteria were enumerated as above by serial micro-dilution. We established the MEC as the concentration where the bacterial viability decreased in 3 log_10_ (99.9%). As shown in Figure 3, the MEC was determined at 4 µg/mL for both PSRe-µL1 (Figure 3A) and PSRe-µL2 (Figure 3B) (** *p* < 0.01, *** *p* < 0.001, Student’s *t*-test comparing to control with 0 µg/mL). The MEC of 4 µg/mL of PSRe-µL1 or PSRe-µL2 were used to determine the lethality time on 1 × 10^7^ CFU/mL of each bacterial strain in ca-MH broth. The mix was exposed to 17 µW/cm^2^ of a white LED light for 5, 15, 30, 60, and 120 min. Control wells with bacteria without PS were also included. For each time, viable bacteria were enumerated by serial micro-dilution and colony counted in ca-MH agar. We established the lethality when the bacterial viability decreased in 3 log_10_ (99.9%). As seen in Figure 3C, although PSRe-µL1 produced a significant reduction after 30 min of incubation (*p* = 0.013, compared to time 0), a 3log_10_ reduction was observed after 60 min of light exposure (*p* < 0.01, compared to time 0). In comparison, the PSRe-µL2 significantly (*p* < 0.01) reduced in 3log_10_ the bacterial viability after 30 min (Figure 3D).

### 3.4. Increased Photodynamic Effect of PSRe-µL1 and PSRe-µL2 in Combination with Imipenem

The desired quality for photosensitizer compounds is to be used as adjunctive therapy with antibiotics. The improvement in combined therapy of the compounds PSRe-µL1 and PSRe-µL2 with imipenem was used to verify their usefulness in eradicating carbapenemase-producing *K. pneumoniae*. The strains of *K. pneumoniae* susceptible to carbapenem KPC^−^ (KPPR1), and the resistant strain KPC^+^ (ST258), were exposed to the preparation of 4 µg/mL of imipenem mixed in aqueous solution with 4 µg/mL of each PSRe-µL1, PSRe-µL2 (corresponding to their MECs), or the control compound PS-Ru [29]. Additionally, control bacteria exposure to light but without imipenem were included. As expected, when mixed with imipenem, the PSRe-µL1 and PSRe-µL2 compounds showed a significantly (*** *p* < 0.001) increased effect over bacterial viability, increasing from 3 to 6log_10_ the bactericidal effect for the KPC**^+^** strain (Figure 4). As seen previously, this behavior was not observed when combining the imipenem with the PS-Ru control compound, keeping the 3log_10_ inhibitory effect (*p* < 0.05) [29].

### 3.5. Antimicrobial Photodynamic Inhibition of the PSRe-µL1 and PSRe-µL2 Compounds over Clinical Isolates

We have in our laboratory a collection of 118 clinical isolates of *K. pneumoniae* from patients who had an active infection [42]. We used these isolates to verify the photodynamic activity of the PSRe-µL1 and PSRe-µL2 compounds on community bacteria and compared them with PS-Ru positive control [49]. As seen in Figure 5, photodynamic treatment with 4 µg/mL PSRe-µL1 or PSRe-µL2 significantly (*** *p* < 0.001; compared to untreated control) inhibits bacterial growth > 3log_10_ (>99.9%) of clinical isolates of *K. pneumoniae*. The results show that the bactericidal effect produced by PSRe-µL1 and PSRe-µL2 is light-dependent (ns = *p* > 0.05; compared to the untreated control). Those results are comparable with the positive control PS-Ru (4 µg/mL) (*** *p* < 0.001). Because this collection of strains has been characterized by its resistance profile and presents 66 ESBL-producing isolates, we analyzed whether a synergic effect occurred with cefotaxime. It was first determined whether the combined treatment with Cfx increases the inhibition of bacterial growth of aPDI with PSRe-µL1 or PSRe-µL2. The clinical isolates were exposed to the preparation of 4 µg/mL of cefotaxime with 4 µg/mL of PSRe-µL1 or PSRe-µL2. As expected, the PSs compounds mixed with cefotaxime significantly (*** *p* < 0.001) increased the bactericidal effect on the clinical isolate population from 3 to 6log_10_ reduction (Figure 5). No significantly increased inhibitory effect was observed for the PS-Ru control combined with cefotaxime (ns *p* > 0.05).

### 3.6. Synergism between aPDI with Cefotaxime and FIC Index Determination

The fractional inhibitory concentration (FIC) index was determined to verify when the PS and Cfx combination increases the bactericidal activity synergistically or additively. We used the MEC determination for the photosensitizer compounds, and the results of the mixture with antibiotics were tabulated as the MIC for simplicity. The set of 66 ESBL-producing clinical isolates of *K. pneumoniae* [42] were mixed with 4 µg/mL PSRe-µL1, PSRe-µL2, or PS-Ru and added to a serial log2 dilutions of Cfx (from 0 to 32 µg/mL). The mixes were incubated for 1 h for aPDI or in the dark, and the Cfx-MIC was determined 16–20 h after. As seen in Figure 6A, compared to the aPDI untreated group, a significant reduction (*** *p* < 0.001) from 8 µg/mL (8–8) to 0.17 µg/mL (0.09–0.34) with PSRe-µL1 and 0.23 µg/mL (0.15–0.41) with PSRe-µL2 on Cfx-MIC was observed (Table 2). In addition, the combined treatment also reduced the PSRe-µL1-MEC from 4 to 0.5 µg/mL and the PSRe-µL2 MEC from 4 to 0.5 µg/mL (Figure 6B, Table 2). Both PSRe-µL1 and PSRe-µL2 compounds produced a significant change in Cfx-susceptibility with an FIC index of 0.15 for (Table 2). Figure 6B and Table 2 shows that the control compound, PS-Ru, did not significantly change Cfx-susceptibility and shows an FIC index of 1.58. Given that synergy is defined with an FIC index ≤ 0.5 [44], the increase in the inhibitory effect shown by the combination of PSRe-µL1 or PSRe-µL2 with Cfx is synergistic and not additive.

### 3.7. The Cytotoxic Effect of PSRe-µL1 and PSRe-µL2 on Mammalian Cells

The PSRe-µL1 and PSRe-µL2 compounds were tested in vitro and found to be secure for mammalian cells. This work tested the intrinsic cytotoxicity (dark cytotoxicity) and light-dependent cytotoxicity of the PSs compounds in the human HEp-2 and HEK293 cell lines. The trypan blue exclusion technique allowed us to determine cell death of 500,000 cells in the presence of 4 µg/mL PSRe-µL1 or PSRe-µL2. The cells were incubated with PSs for 1 h in the dark or activated with 17 µW/cm^2^ of white LED light when the PSs were removed. Fresh medium DMEM with 10% FBS without PS was replaced, and cells were incubated 24 h more in the dark at 37 °C in a 5% CO_2_ atmosphere. As shown in Figure 7, when HEp-2 and HEK293T cells were exposed to 4 µg/mL of PSRe-µL1 or PSRe-µL2 in the dark, no significant reduction in the cell survival was observed (ns = *p* > 0.05, Student’s *t*-test, comparing treated cells with the control). Similarly, the light exposition induced no significant cell death for HEp-2 cells, although a slight (12.5 ± 5%) but significant (*p* > 0.05) reduction in cell viability of PSRe-µL1 over HEK293T cells were shown.

## 4. Discussion

Previously, we reported photosensitizing compounds based on Ir (III), which exhibited absorption processes close to 400 nm wavelength [29]. Therefore, although those compounds are microbicide in vitro, they may be more challenging to access exciting light into the tissues [11]. In this sense, this work tested Re(I) PS compounds that exhibit absorption processes at longer wavelengths, improving the possibility to access excited states into the tissues. Considering the sustained decrease in antibiotic options, these SPs could greatly complement the treatment of infections with MDR microorganisms such as *K. pneumoniae* [1,3]. In this context, using aPDI as complementary therapy becomes viable due to their usefulness as a rescue therapy for infections with MDR bacteria and reverse resistance to antibiotics of choice, reducing the spread of MDR strains [50].

The effective microbial activity of the PSRe-µL1 and PSRe-µL2 photosensitizers rests on the photophysical properties provided by their chemical structure. Although these compounds are neutral (do not have positive charges in the coordination sphere), they can easily polarize by absorption of light, as described in a previous report [41]. This characteristic may improve their molecular proximity to the *K. pneumoniae* cell envelope. Additionally, by exciting these high-rate light-absorbing dark compounds, they can produce high-energy triplet states, which promote energy transfer to molecular oxygen [51]. At first glance, the antimicrobial effect observed with PSRe-µL1 and PSRe-µL2 can be explained by access to a triplet state, as reported [41]. It will then depend on the nature of the Re (I) complex; the excited state responsible for the generation of the ROS producing photooxidative stress could be states ^3^MLCT, ^3^LC, or a mixture of both states [52]. Although aPDI performance is similar for both PSs, the PSRe-µL2 requires less time exposition than PSRe-µL1 to get the same effect. This difference could be associated with the higher coefficients of molar extinction presented for the PSRe-µL2 compound. The difference in the chemical structure of the L2 polypyridine ligand involving more complex hydrocarbon chains may be related [52]. A more extensive photophysical characterization should be carried out to corroborate these hypotheses, such as determining the bacterial cell envelope damage by TEM or the production of reactive oxygen species.

At concentrations as low as 4 µg/mL, both the PSRe-µL1 and the PSRe-µL2 compounds showed an effective aPDI activity, inhibiting the growth of *K. pneumoniae* (>3log_10_). These results are comparable to using cyclometalated Ir (III) complexes as aPDI, as previously reported [23,29]. Similar values have been reported for other PS compounds against Gram-negative bacteria [53,54,55,56]. When combined with imipenem, the PSRe-µL1 and PSRe-µL2 compounds produced a similar effect, shown previously by Ir(III) based cyclometalated compounds [29] and other photosensitizers such as rose bengal for *Acinetobacter baumannii* [57]. Similarly, alternative compounds such as anti-biofilm peptides mixed with conventional antibiotics reported good antimicrobial activity in an in vivo model [58]. This behavior could be related, as was mentioned before, to the external chemical structures of the polypyridine ligands. In the Re(I)-PSRs, polypyridine ligands have heteroatoms, allowing dipole–dipole interactions that may weaken the bacterial cell envelope facilitating the β-lactam action. The synergism shown by these compounds with β-lactams suggests that they damage the bacterial cell wall structure. Oxidative stress is known to modify the membrane permeability of *K. pneumoniae* [59]. However, at the moment, we have not carried out experiments, such as transmission microscopy, to confirm this idea. Our results also show the capacity of the Re(I)-PSRs compounds to inhibit the bacterial growth of a collection of 118 clinical isolates of *K. pneumoniae*. We used this collection of strains because its antibiotic resistance is characterized and presents a well-established sub-population of ESBL-producing bacteria [42]. The aPDI was not only effective; the FIC also demonstrates the synergic combination with Cfx over the total population and a significant reduction in Cfx-MIC in an ESBL-producers subpopulation. The synergistic effect exhibited by these compounds is one of their most remarkable qualities. The use of photodynamic therapy may resolve the lost susceptibility of MDR bacteria. Therefore, photodynamic therapy combined with more conventional antibiotics avoids the use of rescue therapy [50,57]. Similar to Cfx-MIC being reduced in combination, PSs-MEC was also reduced, indicating that lower concentration antibiotic/PS regimens could be used effectively.

The antimicrobial activity of the PSRe-µL1 and PSRe-µL2 is photodynamic and, therefore, its activation is dependent on light. The dependence on light suggests that the PS compounds themselves and in the dark are not toxic, as the results with mammalian cells show us. The slight but significant light-dependent cytotoxicity exhibited by PSRe-µL1 on HEK293T cells reinforces the low intrinsic toxicity. However, it may imply potential damage to the host tissues that must be considered and prevented with a better characterization of the exposure times and concentration of the compound. Furthermore, the cytotoxicity shown by the PSRe-µL1 and PSRe-µL2 compounds is of low significance compared to that shown by the antitumor photosensitizer compounds [60]. However, these compounds must be tested in vivo to establish whether a significant cytotoxic effect may arise, such as an anaphylactic reaction [61]. A murine model may then probe if these compounds could treat infectious diseases in vivo.

For now, it is difficult to accurately calculate the dose of light necessary to activate these PSRe compounds fully; however, we observed in vitro that with photon flux as low as 17µW/cm^2^ of white light, they are bactericidal. Compounds with optimum absorbance at higher wavelengths ranging from 450–750 nm would improve the exposure of PS to light into the tissues, but being less energetic, the PSs must have a triplet excited state of easy access to promote energy transfer [62]. The dark color of PSRe-µL1 and PSRe-µL2 may imply that light of different wavelengths can also be absorbed by this Re-PS, increasing the possibility for it to be excited into the tissues [51]. Therefore, we need to characterize its antimicrobial activity better when activated with defined wavelengths before starting in vivo studies in infection models [63].

## 5. Conclusions

The present study shows that the increased MICs of resistant bacteria could be reverted by aPDI, turning resistant strains into susceptible ones. Thus, aPDI would effectively treat MDR bacteria, greatly complementing antibiotic therapy. Furthermore, therapeutic regimens with lower doses of antibiotics and PS can be used effectively due to the synergistic effect. The requirement for lower doses of antibiotics will help reduce the generation of resistance. In this work, two bimetallic Re(I) compounds were tested as PSs to be used in the aPDI. According to their UV-vis absorption characterization, it was identified that the absorptions at lower energies occur at wavelengths higher than 450 nm, improving its use in infections compared to the PSs based on Ir(III), previously studied. The dark quality in both powder solids and liquid solutions of the Re(I) compounds may imply that light of a wide wavelength range can be absorbed [51], increasing the possibility of it being excited into the tissues. Although its low excitation at more penetrating wavelengths, during infections of internal organs such as UTI, optical fibers can be used through a catheter to deliver the required light dose [64,65]. Furthermore, the bactericidal activity of these PSs compounds occurred at similar concentrations to those used in antibiotic therapies (CLSI 2017). At these concentrations, the PSs showed no dark cytotoxicity or low light-dependent toxicity on mammalian cells.

## Figures and Tables

**Figure 1 pharmaceutics-13-01889-f001:**
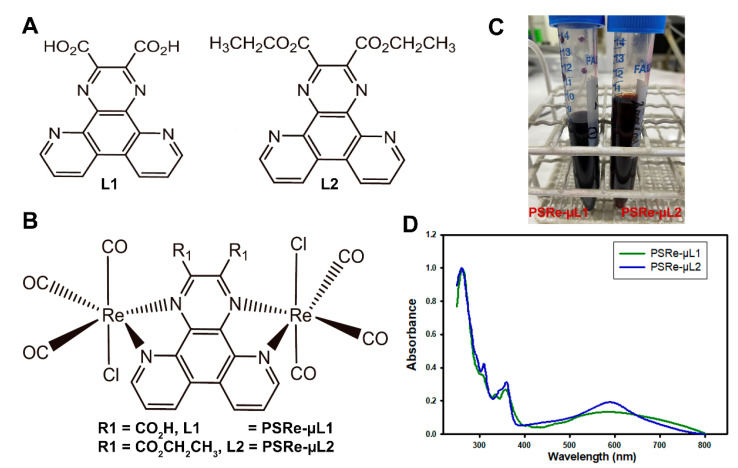
Chemical structure of the Re(I) bimetallic photosensitizer compounds and their L ligands and absorption spectra. The chemical structure of Re(I) complexes show the L1 and L2 ligands (**A**) and the (Re(CO)_3_Cl)_2_µ-N^N, whose replacement of R_1_ results in the PSRe-µL1 and PSRe-µL2 compounds (**B**). Macroscopic appearance of PSRe-µL1 and PSRe-µL2 compounds concentrated in solution at 2 mg/mL (**C**). The absorption spectra for the PSRe-µL1 and PSRe-µL2 compounds in acetonitrile solution (**D**).

**Figure 2 pharmaceutics-13-01889-f002:**
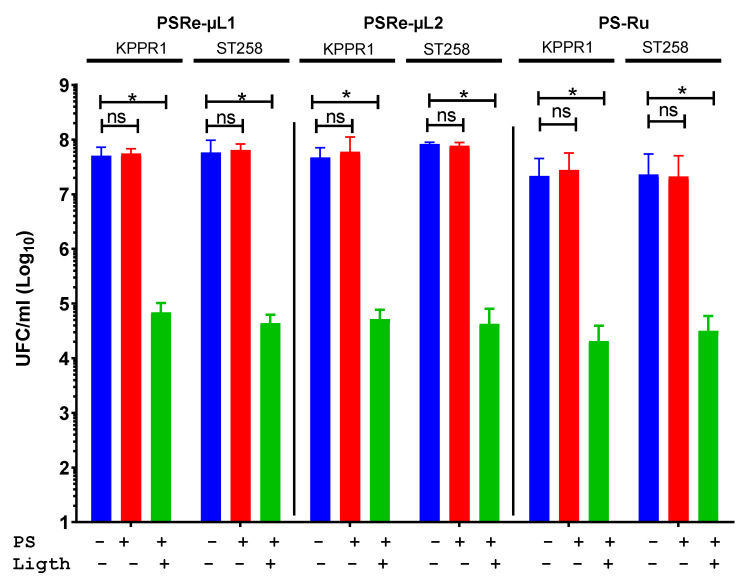
The Photodynamic antimicrobial inhibition of PSRe-µL1 and PSRe-µL2 compounds. The imipenem-sensitive strain (KPPR1) and the imipenem resistant strain (ST258) of *K. pneumoniae* were used at a 1 × 10^7^ CFU/mL and mixed in triplicate with 16 µg/mL of PSRe-µL1, PSRe-µL2, or PS-Ru compounds. The mix was incubated for 1 h at 17 µW/cm^2^ with light for aPDI (green bars) or in the dark for controls (red bars). The results were compared to control of bacteria not combined with the PSs (blue bars). Colony count enumerated viable bacteria on ca-MH agar after serial micro-dilution. The CFU/mL values are presented as means ± SD on a log_10_ scale. The + and − signs indicate the presence or absence of a compound or condition. Not significant (ns) *p* > 0.05 by Student’s *t*-test among bacteria treated with PS without light compared to untreated control bacteria; * *p* < 0.05 by Student’s *t*-test among bacteria treated with PS exposed to light compared to untreated control bacteria.

**Figure 3 pharmaceutics-13-01889-f003:**
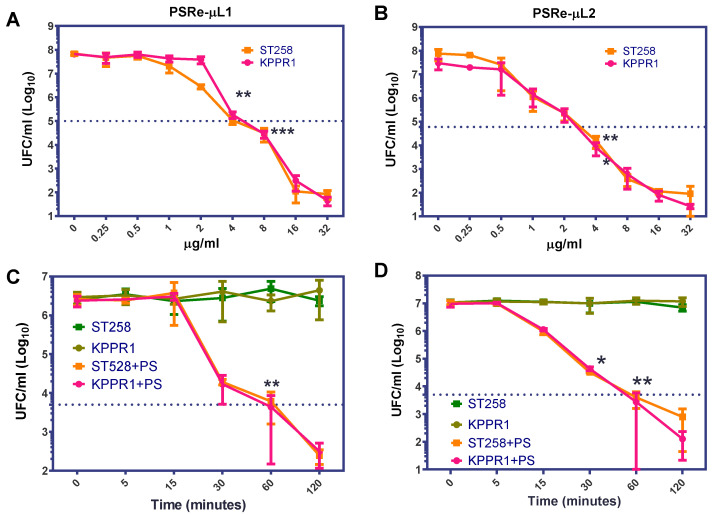
Determination of minimum effective concentration and time lethality. The minimum effective concentration (MEC) and lethality of the PSRe-µL1 and PSRe-µL2 were determined using the imipenem-sensitive (KPPR1) and the imipenem resistant (ST258) strains of *K. pneumoniae*. For MEC determination, bacteria at 1 × 10^7^ CFU/mL were incubated with increasing concentrations (0–32 µg/mL) of the compounds PSRe-µL1 (**A**) or PSRe-µL2 (**B**) and exposed for 1 h to 17 µW/cm^2^ of white LED light. The time lethality was determined, mixing the bacteria with 4 µg/mL of PSRe-µL1 (**C**) or PSRe-µL2 (**D**), and exposure for increasing times (5, 15, 30, 60, and 120 min) to 17 µW/cm^2^ of white LED light. Colony count enumerated viable bacteria on ca-MH agar after serial-microdilution. The CFU/mL values are presented as means ± SD on a log_10_ scale (* *p* < 0.05, ** *p* < 0.01, *** *p* < 0.001 by Student’s *t*-test among bacteria treated with PS exposed to light compared to untreated control bacteria).

**Figure 4 pharmaceutics-13-01889-f004:**
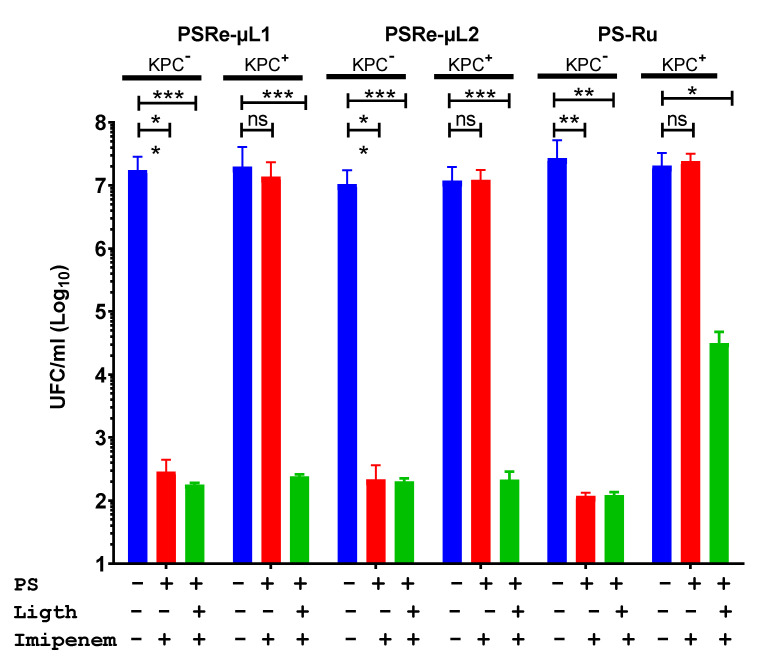
Increased photodynamic effect with imipenem. The effect of carbapenem on the PSRe-µL1 and PSRe-µL2 activity was determined in the strains of *K. pneumoniae* sensitive (KPPR1) or resistant (ST258) to imipenem. The bacteria at 1 × 10^7^ UFC/mL were exposed to a mixture of 4 µg/mL of imipenem and with 4 µg/mL of each PS or the control PS-Ru. For aPDI treatment, the mixes were incubated for 1 h at 17 µW/cm^2^ of light (green bars) or in the dark (red bars). Control also includes bacteria not treated (blue bars). Colony count enumerated viable bacteria on ca-MH agar after serial microdilution. The CFU/mL values are presented as means ± SD on a log_10_ scale. The + and - signs indicate the presence or absence of a compound or condition. Not significant (ns) *p* > 0.05 by Student’s *t*-test among bacteria treated with PS + imipenem without light compared to untreated control bacteria; * *p* < 0.05, ** *p* < 0.01 *** *p* < 0.001 by Student’s *t*-test among bacteria treated with PS + imipenem exposed to light compared to untreated control bacteria.

**Figure 5 pharmaceutics-13-01889-f005:**
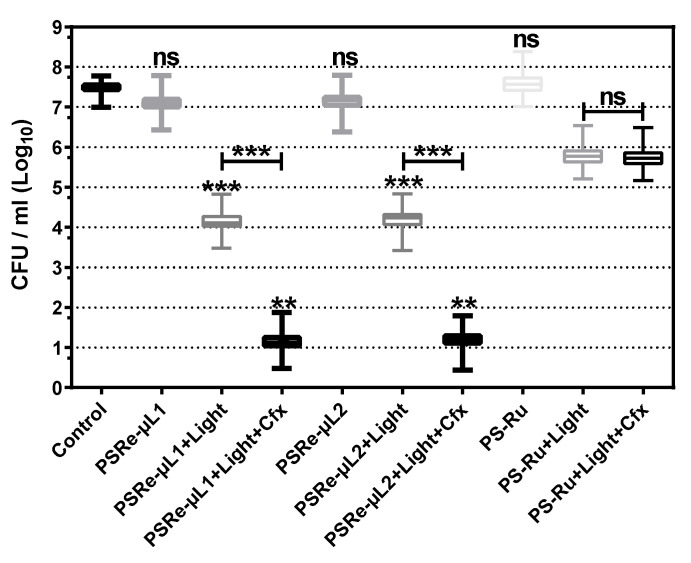
Antimicrobial photodynamic inactivation of clinical isolates of *K. pneumoniae*. (**A**) Growth inhibition of 118 clinical isolates of *K. pneumoniae* subjected to antimicrobial photodynamic inactivation (aPDI) with PSRe-µL1 and PSRe-µL2 compounds compared to control, PS-Ru. The bacteria were utilized at 1 × 10^7^ CFU/mL and mixed in triplicate with 4 µg/mL of PSRe-µL1 and PSRe-µL2 compounds. For the aPDI, the mixture of bacteria with the PSs was exposed for 1 h at 17 µW/cm^2^ of white light. As a control, bacteria combined with the PSs not exposed to light and bacteria not combined with the control PS-Ru were included. Colony count enumerated of viable bacteria on ca-MH agar after serial micro-dilution. The CFUs/mL values are presented as means ± SD on a log_10_ scale. Not significant (ns) *p* > 0.05 by Student’s *t*-test among treated bacteria compared to control; ** *p* < 0.01, *** *p* < 0.001 by Student’s *t*-test among treated bacteria compared to control.

**Figure 6 pharmaceutics-13-01889-f006:**
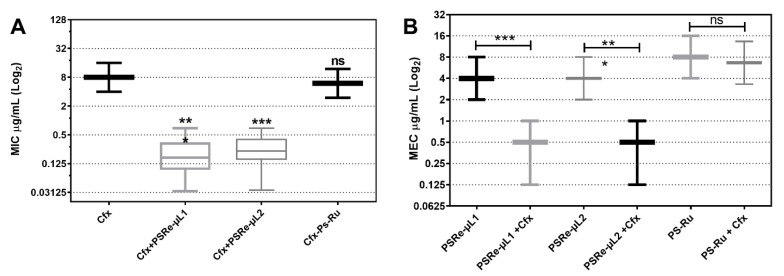
Determination of FIC index using combined photodynamic inactivation and cefotaxime antibiotic. A population of 66 ESBL-producing clinical isolates of *K. pneumoniae* was used to determine the modification on Cfx-MIC by the PSs and the modification in PSs-MECs by cefotaxime. To determine the Cfx-MIC modification, 1 × 10^7^ CFU/mL of bacteria were mixed in triplicate with 4 µg/mL of PSRe-µL1, PSRe-µL2, or PS-Ru. The mix was added to serial dilutions of Cfx and incubated for 1 h with 17 µW/cm^2^ of white light or in the dark. The Cfx-MIC was then determined after 16–20 h at 37 °C in the dark (**A**). The MEC for PSRe-µL1 and PSRe-µL2 were determined in combination with 4 µg/mL of Cfx, performed in triplicate in ca-MH agar for 16–20 h (**B**). The MIC values are presented as median ± SD of µg/mL on a log_2_ scale. Not significant (ns) *p* > 0.05 by Student’s *t*-test among treated bacteria compared to control; * *p* < 0.05, ** *p* < 0.01, *** *p* < 0.001 by Student’s *t*-test among treated bacteria compared to control.

**Figure 7 pharmaceutics-13-01889-f007:**
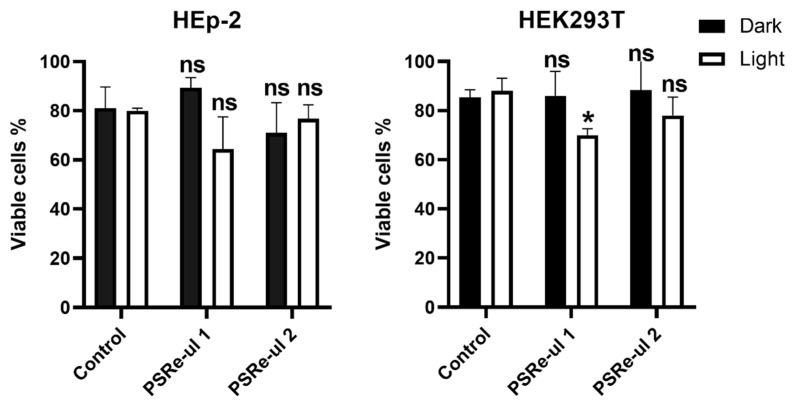
Cytotoxicity of the PSRe-µL1 and PSRe-µL2 compounds. Survival of 5 × 10^5^ cells of HEp-2 and HEK293 human cell lines exposed to 4µg/mL of each PSRe-µL1 and PSRe-µL2 compounds were normalized to control untreated cells and expressed as a percentage of dead cells. Dark cytotoxicity was evaluated with cells exposed to the compound but not activated with light. Light-dependent cytotoxicity was evaluated, exposing the cells for 1 h at 17 µW/cm^2^ of white LED light. Not significant (ns) *p* > 0.05, by Student’s *t*-test between cells exposed to each PS compared to control cells; * *p* < 0.05 by Student’s *t*-test cells exposed to each PS compared to control cells.

**Table 1 pharmaceutics-13-01889-t001:** Photophysical and electrochemical evaluation of photosensitizers.

Compounds	λ_abs_/nm	Area (500–700 nm)	E_ox_1/V	E_ox_2/V	(ε/M^−1^cm^−1^)
PSRe-µL1	358/587	16.28	1.51	1.65	3384
PSRe-µL2	360/588	14.39	1.70		5600
PS-Ru	550	0.83	1.29 *		600 *

* Extracted from Campagna et al., 2007 [47]. λ_abs_, wavelength. Area, the area under the curve. E_ox_, oxidation state. ε, molar absorption.

**Table 2 pharmaceutics-13-01889-t002:** FIC index calculation.

Compounds	^a^ MIC (µg/mL)	MIC Combined (µg/mL)	FIC	FIC Index
Cfx	8.00			
PSRe-µL1	4.00	0.17	0.02	0.15
PSRe-µL1 *		0.50	0.13	
PSRe-µL2	4.00	0.23	0.03	0.15
PSRe-µL2 *		0.50	0.13	
PS-Ru	8.00	6.00	0.75	1.58
PS-Ru *		6.67	0.83	

^a^ MIC values are the median for the ESBL-producing *K. pneumoniae*, *n* = 66. * The PS-MEC values modified by Cfx.

## Data Availability

The data presented in this study are available on request from the corresponding author. The data are not publicly available because they are confidential data of patients protected by the informed consent protocol.

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
