# Peer review of "Effective Treatment against ESBL-Producing Klebsiella pneumoniae through Synergism of the Photodynamic Activity of Re (I) Compounds with Beta-Lactams"

_pharmaceutics, 2021, doi:10.3390/pharmaceutics13111889_

Round 1

Reviewer 1 Report

The present work describes the antibacterial effect of two new photosensitizers (PSRe-μL1 and PSRe-μL2) against a multidrug resistant strain. The introduction gives the necessary information, and the study design is appropriate to the objectives of the work, concerning antibacterial effect. Authors show that the photosensitizers have bactericidal effect that is light-dependent. These are promising results.

However, when testing the cytotoxic effect of the photosensitizers, authors incubate the cells with photosensitizers in the dark, and not in the same conditions as in previous assays. As shown in Figure 2, in the assays with bacteria, the photosensitizers present low dark toxicity (e.g. figure 2), thus it is also expected low dark cytotoxicity (in cell lines). What I would like to know is why these experiments were not done under light exposure, given that the toxicity of these compounds is activated by light.

Also, as made to the bacteria (fig 2) the cytotoxic effect should have been done with and without irradiation in order to compare the toxicity of compounds in the absence and presence of irradiation.

Thus, in Line 448, authors mention that these photosensitizers are not cytotoxic. This only could be stated if these compounds have been tested under light irradiation. I would like to see the results of cell viability of cells exposed to these compounds and subjected to the same irradiation as that used in the bacteria assays (17μW/cm2 of white light).

Minor points

Line 147, the source of the cell lines should be added.

Line 149, please clarify if 2x10Ù7 cells are the density (cells/mL) or the number of cells/well.

Line 202, please correct the units

Author Response

Reviewer N1

Comments and Suggestions for Authors

The present work describes the antibacterial effect of two new photosensitizers (PSRe-μL1 and PSRe-μL2) against a multidrug resistant strain. The introduction gives the necessary information, and the study design is appropriate to the objectives of the work, concerning antibacterial effect. Authors show that the photosensitizers have bactericidal effect that is light-dependent. These are promising results.

However, when testing the cytotoxic effect of the photosensitizers, authors incubate the cells with photosensitizers in the dark, and not in the same conditions as in previous assays. As shown in Figure 2, in the assays with bacteria, the photosensitizers present low dark toxicity (e.g. figure 2), thus it is also expected low dark cytotoxicity (in cell lines). What I would like to know is why these experiments were not done under light exposure, given that the toxicity of these compounds is activated by light.

Also, as made to the bacteria (fig 2) the cytotoxic effect should have been done with and without irradiation in order to compare the toxicity of compounds in the absence and presence of irradiation.

Thus, in Line 448, authors mention that these photosensitizers are not cytotoxic. This only could be stated if these compounds have been tested under light irradiation. I would like to see the results of cell viability of cells exposed to these compounds and subjected to the same irradiation as that used in the bacteria assays (17μW/cm2 of white light).

Answer. As the reviewer suggested, cytotoxicity experiments were performed under irradiation using the same conditions and these results have been incorporated in both results and discussion sections.

Minor points

Line 147, the source of the cell lines should be added.

Answer: As the reviewer requested, the cell lines source was specified.

Line 149, please clarify if 2x10Ù7 cells are the density (cells/mL) or the number of cells/well.

Answer: As the reviewer requested, the cell number in culture are indicated.

Line 202, please correct the units.

Answer: As the reviewer requested, the units were corrected.

Reviewer 2 Report

This article describes the antimicrobial photodynamic (aPDI) activity of two rhenium(I) complexes on two strains of multidrug-resistant bacterium Klebsiella pneumoniae. Given the growing danger that such bacteria pose to health due to insensitivity to antibiotics, this kind of work is very significant and worth publishing. However, there are a number of serious issues that need to be addressed:

1) In the Lines 108-110 it says that the structures and purities of the compounds were confirmed by NMR, FTIR and elemental analysis. Why is this stated here when there are no such data presented in the Results section? Also, the synthesis is not mentioned at all in the discussion where at least the concordance of these data with the literature should be confirmed.

2) How were MIC values determined? This should be clearly described in the Materials and Methods section.

3) What was the time of incubation for the PSs before photoactivation in all aPDI studies? If bacteria were irradiated immediately after addition of the PS, this should be stated.

3) Please explain why was 1 hour of irradiation chosen and such a small white light fluence of 17 uW/cm2. Adding singlet oxygen/ROS measurements for both PSs with the same light conditions is strongly recommended as it would give a stronger evidence for aPDI.

4) Rhenium(CO)3 core is known to exert antibiotic activity without light (see Chem.Eur.J.2020, 26, 2852-2858). Therefore, it is important to obtain MIC values for both PSs without light (‘dark’ toxicity) and to compare it with MIC values obtained with light.

5) Line 185 (Results) says that the PSs were tested by cyclic voltammetry. Why is not then cyclic voltammetry described in the Materials and Methods section?

6) Why was PS-Ru used as a positive control? It was not really discussed anywhere in the manuscript. Maybe it would be also useful to include a structure at page 5.

7) Eox1/V in the Table 1 for PS-Ru was marked as extracted from the Ref. 42. What about other electrochemical properties in the same Table, for PSRe-uL1 and PSRe-uL2 – they seem to be the same as in Ref. 35. This should be properly attributed (was it measured or taken from literature?). Also, the title of the Table 1 is not complete as it includes electrochemical properties as well as photophysical.

8) In Figures 2, 4 and 5 it should say just CFU/mL on the Y-axis, otherwise if it is a log10 then the numbers on the scale are wrong.

9) Why 4 MIC values (16 ug/mL) of two PSs were used? What was the concentration of PS-Ru in the same experiment?

10) Line 236, subsection title 3.3. says determination of the MEC, but there are only MIC values given in the work.

11) Line 247 – this sounds as MBC (minimal bactericidal concentration) not MIC.

12) It should be discussed why FIC index was calculated only for aPDI with cefotaxime and not for imipenem. Based on what is the synergy claimed for imipenem studies?

13) In Figure 4, again concentration of PS-Ru is unknown.

14) In the Section 3.5. the text does not mention Cfx, but in the Table 5, results with Cfx are shown. Also, in the Figure 5 description, Cfx is not mentioned and there is no concentration of PS-Ru.

15) The phototoxic effect of the PSs should also be evaluated on mammalian cells with the same light conditions as for aPDI studies. The light in aPDI would presumably be applied at the infection site, and in that case some mammalian cells would also be affected so it is important to know how much.

16) Lines 245 and458 – 17 um/cm2 is fluence rate, not light dose.

17) Line 406 – aPDI throughout the paper is very similar for both PSs, thus the statement about significantly higher aPDI for PS1 than PS2 is not correct based on all the given results.

18) Line 410 – PS1 and PS2 differ in PS1 having carboxylic groups (thus it is more polar), and PS2 has ester groups. How can ethyl group in PS2 ester promote dipole-dipole interactions as stated? Differences in aPDI effect between these two PSs should be better explained and supported by literature data or more evidence should be provided to support present claims.

19) Panchromatic behaviour was not at all investigated in this work, thus it should not be speculated in the Conclusions.

Author Response

Reviewer N2

Comments and Suggestions for Authors

This article describes the antimicrobial photodynamic (aPDI) activity of two rhenium(I) complexes on two strains of multidrug-resistant bacterium Klebsiella pneumoniae. Given the growing danger that such bacteria pose to health due to insensitivity to antibiotics, this kind of work is very significant and worth publishing. However, there are a number of serious issues that need to be addressed:

As the reviewers suggested, cytotoxicity experiments were performed under irradiation using the same conditions and these results have been incorporated in both results and discussion sections.

  • In the Lines 108-110 it says that the structures and purities of the compounds were confirmed by NMR, FTIR and elemental analysis. Why is this stated here when there are no such data presented in the Results section? Also, the synthesis is not mentioned at all in the discussion where at least the concordance of these data with the literature should be confirmed.

Answer. We appreciate the comments of the reviewer; however, the synthesis of these compounds and structural characterizations are beyond this manuscript scope, and our group previously published them in detail. Then we feel that repeating those data would be redundant. However, we include information that we think is relevant to understanding the work done in this manuscript, namely: chemical structure of compounds, absortion spectra and an explanatory lines (yellow highlighted) in the new version of the manuscript.

How were MIC values determined? This should be clearly described in the Materials and Methods section.

Answer. As the reviewer requested, a better explanation of how the MEC was determined for the PS-Ru compound was included.

  • What was the time of incubation for the PSs before photoactivation in all aPDI studies? If bacteria were irradiated immediately after the addition of the PSs, this should be stated.

Answer. As the reviewer requested, the time between the addition of the compound and the exposition was incorporated in the Materials and Methods sections.

  • Please explain why was 1 hour of irradiation chosen and such a small white light fluence of 17 uW/cm2. Adding singlet oxygen/ROS measurements for both PSs with the same light conditions is strongly recommended as it would give a stronger evidence for aPDI.

Answer. We do not have a light source with greater power or the technical capacity to measure singlet oxygen production in our setting. We did a lethality time experiment, and we found 1h exposition shows the best results. Changing our light source will significantly improve our research and reduce the irradiation time; we are working on it. Furthermore, we are also working to quantify ROS production under these conditions.

  • Rhenium(CO)3core is known to exert antibiotic activity without light (see Chem.Eur.J.2020, 26, 2852-2858). Therefore, it is important to obtain MIC values for both PSs without light (‘dark’ toxicity) and to compare it with MIC values obtained with light.

Answer. In our setting at 16 ug/ml, the PSReL1 and PSReL2 compounds were not bactericidal in the absence of light, as can be seen in figure 2. Therefore, we do not consider it necessary to determine the MIC in the dark.

  • Line 185 (Results) says that the PSs were tested by cyclic voltammetry. Why is not then cyclic voltammetry described in the Materials and Methods section?

Answer. The Cyclic voltammetry was incorporated into the materials and methods section as part of the characterization. Also, on line 185, it was paraphrased (in yellow).

  • Why was PS-Ru used as a positive control? It was not really discussed anywhere in the manuscript. Maybe it would be also useful to include a structure at page 5.

Answer. The PS-Ru has been used by other authors and in our setting as a positive control for Photodynamic behavior. The references have been updated in the Introduction section and discussion section.

  • Eox1/V in the Table 1 for PS-Ru was marked as extracted from the Ref. 42. What about other electrochemical properties in the same Table, for PSRe-uL1 and PSRe-uL2 – they seem to be the same as in Ref. 35. This should be properly attributed (was it measured or taken from literature?). Also, the title of the Table 1 is not complete as it includes electrochemical properties as well as photophysical.

Answer. All data in Table 1, except PS-Ru voltammetry, come from our compound characterization study, which was previously published (Ref. 35). Because many other data appear in the previous publication, Table 1 shows only those we consider relevant to understand this work better. The study was better referenced in the text for those who want more specific information.

  • In Figures 2, 4 and 5 it should say just CFU/mL on the Y-axis, otherwise if it is a log10 then the numbers on the scale are wrong.

Answer. As the reviewer requested, the legend in the Y-axis of figures 2, 4, and 5 was corrected.

  • Why 4 MIC values (16 ug/mL) of two PSs were used? What was the concentration of PS-Ru in the same experiment?

Answer. The concentration of 16 ug / mL was used in the first experiment to determine the bactericidal capacity of the compound and its dependence on light, as shown in figure 2. Then, as shown in Figure 3, the minimal effective concentration of PS1 and PS2 was determined at 4 ug/ml. Next, the combination experiments with imipenem and cefotaxime, potency, and cytotoxicity were performed at 4 ug/ml, their PS-MECs.

  • Line 236, subsection title 3.3. says the determination of the MEC, but there are only MIC values given in the work.

Answer. The reviewer pointed out that the minimum concentration results performed for the PSs refer more to a MEC than a MIC. That is why we have replaced all of them in the text.

  • Line 247 – this sounds as MBC (minimal bactericidal concentration) not MIC.

Answer. See the above answer.

  • It should be discussed why FIC index was calculated only for aPDI with cefotaxime and not for imipenem. Based on what is the synergy claimed for imipenem studies?

Answer. As the reviewer suggested, a discussion about the decision to present the FIC index results was incorporated in the discussion section.

  • In Figure 4, again concentration of PS-Ru is unknown.

Answer. As the reviewer suggested, the PS-Ru concentration was included in the material and methods section.

  • In the Section 3.5. the text does not mention Cfx, but in the Table 5, results with Cfx are shown. Also, in the Figure 5 description, Cfx is not mentioned and there is no concentration of PS-Ru.

Answer. As the reviewer suggested, the mention of Cfx presented in figure 5 was moved to section 3.5.

  • The phototoxic effect of the PSs should also be evaluated on mammalian cells with the same light conditions as for aPDI studies. The light in aPDI would presumably be applied at the infection site, and in that case some mammalian cells would also be affected so it is important to know how much.

Answer. As suggested by the reviewer, the cytotoxicity under irradiation conditions was performed for Hep-2 and HEK cells and incorporated into the results and discussion sections.

  • Lines 245 and458 – 17 um/cm2 is fluence rate, not light dose.

Answer. As suggested by the reviewer, the text was modified.

  • Line 406 – aPDI throughout the paper is very similar for both PSs, thus the statement about significantly higher aPDI for PS1 than PS2 is not correct based on all the given results.

Answer. As suggested by the reviewer, the interpretation was modified. The performance difference, although slight, is significant only in one aspect of the characterization, the irradiation time.

  • Line 410 – PS1 and PS2 differ in PS1 having carboxylic groups (thus it is more polar), and PS2 has ester groups. How can ethyl group in PS2 ester promote dipole-dipole interactions as stated? Differences in aPDI effect between these two PSs should be better explained and supported by literature data or more evidence should be provided to support present claims.

Answer. As suggested by the reviewer, those conjectures that are not supported by the results achieved in this work were eliminated.

  • Panchromatic behaviour was not at all investigated in this work, thus it should not be speculated in the Conclusions.

Answer. As requested by the reviewer, the interpretations of the results were modified, and the conjectures about interaction not assessed were removed from the text.

Round 2

Reviewer 1 Report

The authors have answered all the questions and corrected/clarified the manuscript. Figure 7 has been completed with the necessary information, the data is now clearer. In my opinion the manuscript is ready to be published.

Reviewer 2 Report

The manuscript has been improved and I recommend it for publication.